# Bisphenol A Alters the Energy Metabolism of Stromal Cells and Could Promote Bladder Cancer Progression

**DOI:** 10.3390/cancers13215461

**Published:** 2021-10-30

**Authors:** Ève Pellerin, Stéphane Chabaud, Frédéric Pouliot, Martin Pelletier, Stéphane Bolduc

**Affiliations:** 1Centre de Recherche en Organogénèse Expérimentale/LOEX, Regenerative Medicine Division, CHU de Québec-Université Laval Research Center, Québec, QC G1J 1Z4, Canada; eve.pellerin@crchudequebec.ulaval.ca (È.P.); Stephane.Chabaud@crchudequebec.ulaval.ca (S.C.); 2Oncology Division, CHU de Québec-Université Laval Research Center, Québec, QC G1R 2J6, Canada; frederic.pouliot@fmed.ulaval.ca; 3Department of Surgery, Faculty of Medicine, Laval University, Québec, QC G1V 0A6, Canada; 4Infectious and Immune Disease Division, CHU de Québec-Université Laval Research Center, Québec, QC G1V 4G2, Canada; 5Department of Microbiology-Infectious Diseases and Immunology, Faculty of Medicine, Laval University, Québec, QC G1V 0A6, Canada

**Keywords:** bisphenol A, cancer-associated fibroblasts, bladder cancer, metabolism, glycolysis

## Abstract

**Simple Summary:**

Our research brings new insight on the potential impact of bisphenol A on bladder cancer progression. By evaluating the effects of bisphenol A on the stromal environment of bladder cancer, we aimed to demonstrate that this endocrine disruptor could promote bladder cancer invasion through alteration of the energy metabolism of stromal cells, specifically on bladder fibroblasts and cancer-associated fibroblasts. These findings could modify the understanding of bladder cancer since bladder tissue is not recognized as a hormone-sensitive tissue. Consequently, our study suggests that endocrine disruptors, such as bisphenol A, could impact bladder cancer progression.

**Abstract:**

Bisphenol A (BPA) is an endocrine-disrupting molecule used in plastics. Through its release in food and the environment, BPA can be found in humans and is mostly excreted in urine. The bladder is therefore continuously exposed to this compound. BPA can bind to multiple cell receptors involved in proliferation, migration and invasion pathways, and exposure to BPA is associated with cancer progression. Considering the physiological concentrations of BPA in urine, we tested the effect of nanomolar concentrations of BPA on the metabolism of bladder fibroblasts and cancer-associated fibroblasts (CAFs). Our results show that BPA led to a decreased metabolism in fibroblasts, which could alter the extracellular matrix. Furthermore, CAF induction triggered a metabolic switch, similar to the Warburg effect described in cancer cells. Additionally, we demonstrated that nanomolar concentrations of BPA could exacerbate this metabolic switch observed in CAFs via an increased glycolytic metabolism, leading to greater acidification of the extracellular environment. These findings suggest that chronic exposure to BPA could promote cancer progression through an alteration of the metabolism of stromal cells.

## 1. Introduction

Bisphenol A (BPA) is a chemical compound used to produce plastics, such as polycarbonate, polyesters and epoxy resins [1]. It is used in a wide variety of daily products, such as water bottles and food containers. BPA molecules can be released in food and the environment [1], which causes humans and animals to be continuously exposed. Once in the organism, BPA is partially metabolized and excreted, mainly via urine [2]. It is possible to detect measurable concentrations of BPA in more than 90% of urine samples from humans [3]. Due to its molecular structure being similar to that of estrogen, BPA binds to multiple cellular receptors such as estrogen receptors (ERs), the androgen receptor (AR) and the G-protein-coupled estrogen receptor (GPER) [1]. Thus, acting as an endocrine disruptor, BPA affects signaling pathways related to proliferation [4], cell migration [4,5], invasion [5] and apoptosis [6]. BPA exposure is also associated with cancer development, especially for hormone-dependent cancers such as breast [5] and prostate cancer [7].

Bladder tissue is not recognized as a hormone-dependent tissue, but studies have shown that ERs and the AR are involved in bladder cancer initiation and progression [8,9]. Considering the presence of BPA in urine and the presence of steroid receptors in the bladder urothelium, BPA could have a potential role in bladder cancer development [10].

A hallmark of tumors is the alteration of the metabolic profile of cancer cells, known as the Warburg effect, which consists of a metabolic switch from mitochondrial respiration to glycolytic metabolism [11]. Aerobic glycolysis allows cancer cells to synthesize ATP faster, although providing less overall ATP generation, since the production of lactate from glucose is much faster than the complete oxidation of glucose through the mitochondria [12]. Furthermore, the Warburg effect leads to the production of excess biosynthetic intermediates in the cell, allowing it to generate proteins, lipids and nucleotides, therefore supporting the increased proliferation of cancer cells [13]. Additionally, the transformation of glucose to lactate allows the cell to regenerate NAD+, which is essential to maintain glycolysis [11]. From the point of view of the tumor microenvironment (TME), the Warburg effect can be favorable for cancer cells. The increased glycolytic metabolism leads to higher lactate production, therefore decreasing the local pH and provoking acidification of the TME [11]. This acidification can be beneficial for cancer cells; for example, it alters the stroma, which promotes cancer cell invasion and metastasis [14].

The TME is characterized by multiple cell types, including ones from the epithelium and the stroma, that communicate and interact with each other [15]. Among them, cancer-associated fibroblasts (CAFs) play a critical role in cancer progression. Studies by Ringuette-Goulet et al. have shown that invasive bladder cancer cells release TGF-β, which induces normal fibroblasts into CAFs [16]. In turn, CAFs secrete IL-6, which increases the epithelial–mesenchymal transition of non-invasive bladder cancer cells and could allow cell invasion in the stroma [17].

Considering the omnipresence of BPA in urine and its impact as an endocrine disruptor through ERs and the AR, we hypothesized that BPA would impact the metabolism of healthy and cancer-associated bladder fibroblasts, which could promote bladder cancer progression.

## 2. Materials and Methods

### 2.1. Cell Lines

All procedures involving patients were conducted according to the Helsinki Declaration and were approved by the local Research Ethical Committee. Donors’ consent for tissue harvesting was obtained for each specimen, and experimental procedures were performed according to the CHU de Québec guidelines. Human bladder fibroblasts (HBFs) were extracted from a normal human urological tissue biopsy and were cultured as previously described [18]. The HBFs used were all from the same patient (not transformed primary cell line).

HBFs, non-invasive RT4 bladder cancer cells (ATCC HTB-2) and invasive T24 bladder cancer cells (ATCC HTB-4) were used. Cells were grown in culture media composed of a 3:1 mix of the Dulbecco-Vogt modification of Eagle’s medium (DMEM) (Invitrogen, Burlington, ON, Canada) and Ham F12 medium (Invitrogen) supplemented with 5% fetal bovine serum clone II (HyClone, GE Healthcare Life Science, Wauwatosa, WI, USA), 24.3 µg/mL adenine (Corning, Tewksbury, MA, USA), 10 ng/mL epidermal growth factor (Austral Biologicals, San Ramon, CA, USA), 0.212 µg/mL isoproterenol (Sandoz, Boucherville, QC, Canada), 5 µg/mL insulin (Sigma-Aldrich, Oakville, ON, Canada), 0.4 mg/mL hydrocortisone (Calbiochem, San Diego, CA, USA), 100 U/mL penicillin (Sigma-Aldrich) and 25 mg/mL gentamicin (Schering-Plough Canada Inc./Merck, Scarborough, ON, Canada) and incubated at 37 °C with 8% CO_2_. Media were changed three times per week.

### 2.2. CAF Induction

HBFs were seeded in 96-well Seahorse XF cell culture plates (Agilent/Seahorse Bioscience, Chicopee, MA, USA) 72 h before CAF induction to allow cell adhesion and confluence, thus limiting cell proliferation during treatment. RT4 and T24 bladder cancer cells were seeded in 6-well plates and exposed or not to 10^−8^ M BPA (Millipore Sigma, Oakville, ON, Canada) for 72 h. At day 0 of CAF induction, conditioned medium from RT4 or T24 ± BPA was collected and centrifuged at 300× *g* for 10 min and added to HBF cultures. BPA was also directly added in some HBF culture conditions to study the direct exposure. Exposure to conditioned media was maintained for 8 days to ensure CAF induction [16]. HBFs were also directly exposed to 10^−8^ M BPA in parallel to CAF induction, and HBFs without BPA or conditioned media were used as controls (Figure 1). Media were changed three times per week. Bioenergetic parameters were measured on day 8. CAF induction was confirmed by α-smooth muscle actin (α-SMA) expression through FACS analysis, as previously demonstrated [16].

### 2.3. Seahorse Energy Metabolism Measurements

Seahorse XFe96 sensor cartridge plates (Agilent/Seahorse Bioscience) were hydrated the day before the analysis with the XF Calibrant (Agilent/Seahorse Bioscience) and incubated at 37 °C without CO_2_ overnight. Before the energy metabolism measurements, cells were washed and incubated for 1 h with Glyco Stress media or Mito Stress media. Glyco Stress media consisted of XF Base Medium (minimal DMEM) (Agilent/Seahorse Bioscience) supplemented with 2 mM l-glutamine (Wisent Bioproducts Inc., Saint-Jean-Baptiste, QC, Canada). Mito Stress media consisted of XF Base Medium supplemented with 2 mM l-glutamine, 1 mM sodium pyruvate (Wisent Bioproducts Inc.) and 10 mM d-(+)-glucose (Millipore Sigma). The extracellular acidification rate (ECAR), representative of glycolytic metabolism, and the oxygen consumption rate (OCR), representative of mitochondrial respiration, were determined using the XFe Extracellular Flux Analyzer (Agilent/Seahorse Bioscience) [19]. The glycolytic metabolism was established by the sequential injection of 10 mM d-(+)-glucose (Millipore Sigma), 1.5 µM of the ATP synthase inhibitor oligomycin (Cayman Chemical, Ann Arbor, MI, USA) to inhibit mitochondrial respiration and force the cells to maximize their glycolytic capacity and 50 mM 2-deoxy-d-glucose (2-DG) (Alfa Aesar, Ward Hill, MA, USA), a competitive inhibitor of the first step of glycolysis. The mitochondrial respiration was measured by the sequential injection of 1.5 µM of the ATP synthase inhibitor oligomycin (Cayman Chemical), 0.5 µM of the mitochondrial uncoupler trifluoromethoxy carbonylcyanide phenylhydrazone (FCCP) (Cayman Chemical) and a combination of 0.5 µM of the mitochondrial complex I inhibitor rotenone (MP Biomedicals, Santa Ana, CA, USA) and 0.5 µM of the mitochondrial complex III inhibitor antimycin A (Millipore Sigma). The concentrations indicated for each injection represent the final concentrations in the wells. At least three measurement cycles (3 min of mixing + 3 min of measuring) were completed before and after each injection. The OCR and ECAR were calculated using Wave software v2.6 (Agilent/Seahorse Bioscience). Energy metabolism was normalized according to the number of cells using a CyQuant Cell proliferation assay kit (Invitrogen) following the manufacturer’s instructions. The fluorescence of each well was measured at 485 nm/535 nm for 0.1 s using the Victor2 1420 MultiLabel Counter plate reader (Perkin Elmer Life Sciences, Waltham, MA, USA) and Wallac 1420 software (PerkinElmer). The normalization values were calculated from the fluorescence measurements with Microsoft Excel software (Microsoft, Redmond, WA, USA) and applied to the metabolic values. Each experiment included eight replicates (*n* = 8), and each experiment was repeated three times (*N* = 3).

### 2.4. Statistical Analysis

Graphical representation and statistical analyses were performed using GraphPad Prism Software v.9.2 (San Diego, CA, USA). The results are expressed as mean ± standard error of the mean (SEM). Statistical analyses were performed using the unpaired Student’s *t*-test or one-way analysis of variance (ANOVA). Statistical significance was established at *p* < 0.05.

## 3. Results

### 3.1. Healthy Human Bladder Fibroblasts Exhibit Decreased Glycolytic and Mitochondrial Metabolism following Chronic Exposure to BPA

Human bladder fibroblasts (HBFs) were exposed to physiological concentrations of BPA to evaluate the impact of this compound on these cells. In vivo, BPA can reach HBFs through blood vessels nourishing the cells populating the stroma and potentially affects them. HBFs chronically exposed to 10^−8^ M BPA exhibited a generally decreased energy metabolism compared to untreated HBFs (Figure 2A,D). HBFs exposed to BPA demonstrated a significantly reduced basal glycolytic metabolism (Figure 2B), while maximal glycolysis slightly decreased, with a *p*-value established at 0.0528 (Figure 2C). HBFs exposed to BPA also exhibited significantly decreased basal and maximal mitochondrial respiration (Figure 2E,F). Therefore, HBFs chronically exposed to physiological concentrations of BPA are characterized by a reduced glycolytic and mitochondrial oxidative metabolism.

### 3.2. Cancer-Associated Fibroblasts Conditioned by Non-Invasive Bladder Cancer Cells Exhibit a Metabolic Switch, Characterized by a Decreased Mitochondrial Metabolism and Increased Glycolysis, Accentuated by BPA

HBFs were induced into cancer-associated fibroblasts (CAFs) using cell culture media conditioned by non-invasive bladder cancer cells (RT4). Compared to HBFs, CAFs conditioned by RT4 cell medium demonstrated generally increased glycolysis (Figure 3A) and significantly decreased mitochondrial respiration (Figure 3D), leading to a metabolic switch similar to the Warburg effect. When RT4 cells were chronically exposed to 10^−8^ M BPA, CAFs exhibited a significantly increased basal glycolytic capacity (Figure 3B) and maximal glycolytic capacity (Figure 3C). When HBFs were directly exposed to BPA during CAF induction, glycolysis was not affected, and the basal and maximal glycolytic capacities were similar to those of unexposed CAFs. Both the basal and maximal levels of mitochondrial respiration of CAFs remained unchanged when exposed directly or indirectly to BPA (Figure 3E,F). The metabolic switch observed in CAFs was therefore accentuated by chronic exposure of RT4 cells to physiological concentrations of BPA through increased glycolysis.

### 3.3. Cancer-Associated Fibroblasts Conditioned with Invasive Bladder Cancer Cells in the Presence of BPA Exhibit an Increased Glycolytic Metabolism

HBFs were induced into CAFs using cell culture media conditioned by invasive bladder cancer cells (T24). Compared to HBFs, CAFs conditioned by the culture medium of T24 cells did not exhibit a metabolic switch, unlike CAFs conditioned with RT4 culture medium. CAFs conditioned with T24 had similar glycolysis and mitochondrial respiration levels as HBFs (Figure 4A,D). The basal and maximal glycolytic capacities were unchanged by CAF induction (Figure 4B,C). However, CAFs exhibited significantly decreased basal mitochondrial respiration (Figure 4E), but the maximal mitochondrial respiration levels were similar to those of HBFs (Figure 4F). Therefore, CAF induction with T24-conditioned media did not seem to affect energy metabolism. On the other hand, when CAFs were directly or indirectly exposed to 10^−8^ M BPA, they exhibited an increased glycolytic metabolism. Chronic exposure to BPA resulted in a significantly increased basal (Figure 4B) and maximal glycolytic capacity (Figure 4C). Although direct exposure to BPA caused a significant increase in glycolysis, the chronic exposure of T24 to BPA (indirect exposure) resulted in a greater increase in both basal and maximal glycolytic capacities (Figure 4B,C). Chronic exposure to BPA did not seem to affect the mitochondrial respiration of CAFs conditioned with T24. Therefore, our results demonstrated that CAFs conditioned with T24 in the presence of BPA exhibit an increased glycolytic metabolism.

## 4. Discussion

As plastic products become more and more ubiquitous in our environment, studying the effects of the chemical compounds they can release, such as BPA, has become crucial. Growing evidence has shown the pro-tumorigenic effect of BPA [4,5,6] in many cancers, especially hormone-dependent cancers such as breast [5] and prostate cancer [7]. Although bladder tissue is not recognized as a hormone-sensitive tissue, studies have demonstrated the role of hormone receptors in bladder cancer initiation and progression [8,9], as well as their impact on treatment [20] and prognosis [21]. As the impact of BPA on bladder cancer has not yet been reported, the effects of BPA on the stromal environment of bladder cancer were studied through bladder fibroblasts and CAFs.

HBFs chronically exposed to physiological concentrations of BPA exhibited a decreased energy metabolism, characterized by a decreased glycolytic capacity and mitochondrial respiration. The consequences of cell detoxification could partly explain these observations. When cells are exposed to toxic substances, cells will detoxify themselves through ATP-binding cassette (ABC) transporters [22]. ABC transporters use the hydrolysis of adenosine triphosphate (ATP) to export these toxic molecules out of the cells [22], and biochemical studies revealed that up to two ATP molecules can be required to export one molecule of substrate [23]. Thus, the translocation of substrates at the expense of ATP hydrolysis catalyzed by ABC transporters leads to substantially high costs in energy consumption [24]. In this sense, bisphenols, including BPA, have been shown to interact with the ABC transporter breast cancer resistance protein (BCRP; coded by the gene *ABCG2*) [25]. Furthermore, studies have shown that BPA can inhibit metabolizing enzymes [26], thus affecting major metabolic pathways, such as mitochondrial respiration [27]. This inhibition could lead to a decreased availability of substrates for metabolic pathways, resulting in a reduced energy metabolism. Our results have shown that HBFs have a decreased energy metabolism when exposed to BPA, suggesting that fibroblasts could be less functional. This could have substantial consequences on cellular functions, such as extracellular matrix (ECM) production. Fibroblasts are responsible for ECM production, which is crucial for tissue turnover and repair [28]. A decreased energy metabolism in HBFs caused by BPA exposure could affect wound healing and ECM turnover and compromise the bladder wall’s capacity to repair damages or adequately sustain a differentiated urothelium with an optimal barrier function [29,30]. Urine contains toxic substances such as urea and carcinogens [31]. Studies have shown that the exposure of the urothelium to urinary carcinogens is linked to bladder cancer [32]. Therefore, a diminished wound healing capacity could result in the exposure of basal and intermediate urothelial cell layers to these substances and promote bladder cancer initiation through cell damage.

We also studied the impact of BPA on CAFs. First, the induction of HBFs into CAFs resulted in a metabolic switch characterized by an increased glycolytic capacity and decreased mitochondrial respiration, similar to the Warburg effect. Similar to our observation, it was reported that CAFs exhibit a metabolic switch where CAFs favor glycolysis to the detriment of mitochondrial respiration, even in the presence of oxygen [33]. Studies have suggested that TGF-β could have a role in this metabolic switch by increasing aerobic glycolysis, increasing oxidative stress, affecting the mitochondria’s functioning and regulating certain enzymes [33,34,35]. Furthermore, studies by Ringuette-Goulet et al. have shown that invasive bladder cancer cells (T24) secrete more TGF-β than non-invasive bladder cancer cells (RT4) do [16]. Interestingly, TGF-β has been shown to activate the PI3K-Akt-mTOR pathway and promote the accumulation of activating transcription factor 4 (ATF4) in lung fibroblasts, leading to their metabolic reprogramming [36]. Another molecule that could be implicated is IL-6, which could impact the physiological activity of fibroblasts as well as potentialize the effects of TGF-β [37]. However, whether the alterations mediated by BPA in the energy metabolism of stromal cells are linked to TGF-β or IL-6 remains to be demonstrated.

CAF induction leads to increased glycolysis and, consequently, to increased production of lactate. An increased concentration of lactate in the ECM can have multiple consequences. Lactate inhibits immune cells, including monocytes involved in cancer cell elimination [38], which could promote cancer initiation and progression through the inhibition of the immune system [35]. Lactate directly leads to the acidification of the extracellular environment, promoting tumor invasion through a reorganization of the matrix [35,39]. Lactate also acts as a chemoattractant molecule and is used as a metabolite for cancer cells [33]. Thus, the increased lactate production following CAF induction allows cancer cells to increase their energy intake through the metabolites provided by CAFs [33]. The increased glycolytic capacity we observed, associated with an increased extracellular acidification rate, concurs with the increased lactate production described in the literature.

Our results on CAFs incubated with BPA showed that the metabolic switch observed in CAFs was accentuated through increased glycolysis when the cell culture medium was conditioned with non-invasive RT4 bladder cancer cells chronically exposed to physiological concentrations of BPA. Considering the impact of ECM acidification, the increased glycolysis resulting from BPA exposure could lead to higher lactate production. Therefore, by enhancing the glycolytic metabolism of CAFs, BPA could accentuate the consequences of lactate in the ECM, resulting in a more important inhibition of the immune cells, a higher energy intake for cancer cells and a critical matrix remodeling. By affecting the glycolytic metabolism of CAFs, BPA could promote the invasion of non-invasive cancer cells through alteration of the ECM and therefore support bladder cancer progression through stromal cells.

Compared to CAFs conditioned with medium from non-invasive RT4 cells, CAFs conditioned with medium from invasive T24 cells did not exhibit a metabolic switch similar to the Warburg effect. The absence of a metabolic switch for CAFs conditioned with medium from T24 cells could partially be explained by the fact that T24 cells are already invasive cancer cells and could therefore be less likely to need CAFs’ contribution to allow cell invasion. Furthermore, the limited changes in the metabolism of CAFs conditioned with medium from T24 cells could be explained by the simplicity of the 2D model, which does not include the interactions between cells and ECM proteins. Certain stimuli, such as the compression induced by the tumor mass on the stroma, will induce a metabolic change [40]. However, it is important to note that BPA by itself can induce similar metabolic changes. The presence of BPA in a 3D model could potentially accentuate these changes.

Although chronic exposure to BPA did not affect the mitochondrial respiration of CAFs conditioned by invasive T24 bladder cancer cells, our results demonstrated that BPA induced an increased glycolytic metabolism. As explained earlier, increased glycolysis leads to an augmentation of lactate production and acidification of the ECM. Even if T24 cells are invasive cancer cells, BPA could facilitate cancer cell invasion by altering the glycolytic metabolism of CAFs and lead to metastasis generation. BPA could therefore promote invasion of bladder cancer cells.

The use of an in vitro 2D cell culture is one of the limitations of this study. The use of a 3D bladder cancer model [18] could allow a better physiological representation of the impact of BPA on the stromal environment. A 3D model could allow us to further study the impacts of chronic exposure to BPA on ECM production by bladder fibroblasts as well as ECM alteration through CAF induction and BPA exposure. Another limitation of this study is the inability to obtain a perfectly bisphenol-free control in experimental settings [41]. In fact, external bisphenol contamination is difficult to avoid because of the ubiquity of bisphenol-based plastics in laboratory equipment and scientific instruments and the leaching properties of these compounds [42]. Thus, the bisphenol concentrations we added in our experiments might be slightly underestimated, but we nonetheless observed significant differences between the BPA condition and the negative controls. The impact of BPA alternatives, such as bisphenol S (BPS), on bladder cancer would also be of relevance. BPS is now used in a wide variety of plastic products to progressively replace BPA [43]. Although BPS is supposed to be a safer alternative, growing evidence shows that BPS has endocrine-disrupting capabilities similar to or even stronger than BPA [43,44] and could also promote cancer progression [45].

## 5. Conclusions

Our study has brought a new insight on the potential impact of BPA, an environmental pollutant, on bladder cancer progression with the evaluation of the impact of BPA on the stromal environment of bladder cancer. Using bladder fibroblasts and CAFs, we demonstrated that the energy metabolism of HBFs is negatively affected by BPA, while the exposure of CAFs to BPA could promote bladder cancer progression and invasion through an altered metabolism. The observed effect of BPA on HBFs and CAFs could partially explain the physiopathology of bladder cancer. In fact, bladder cancer is characterized by elevated levels of recurrence where non-invasive bladder cancer can evolve into invasive bladder cancer. Therefore, ubiquitous and continuous exposure to endocrine disruptors, such as BPA, could have an impact at the clinical level and affect patients’ prognosis.

## Figures and Tables

**Figure 1 cancers-13-05461-f001:**
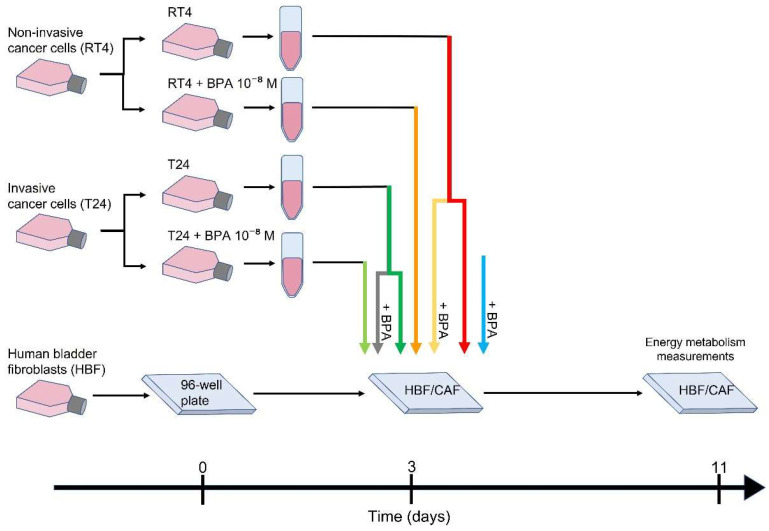
Experimental design. Non-invasive bladder cancer cells (RT4) and invasive bladder cancer cells (T24) were cultured and exposed or not to 10^−8^ M BPA for 72 h to mimic its presence in urine (indirect exposure). Human bladder fibroblasts (HBFs) were seeded in a 96-well Seahorse XF cell culture plate. After 72 h, cancer cell-conditioned media were collected, centrifuged and added to HBFs for induction into cancer-associated fibroblasts (CAFs). Healthy HBFs were used as controls. Furthermore, 10^−8^ M BPA was also added to HBFs and HBF/CAFs conditioned with RT4/T24 media to mimic direct exposure of the stroma to BPA. After 8 days of exposure to conditioned media in the presence or absence of BPA, energy metabolism was measured using the XFe Extracellular Flux Analyzer via glycolysis and mitochondrial respiration.

**Figure 2 cancers-13-05461-f002:**
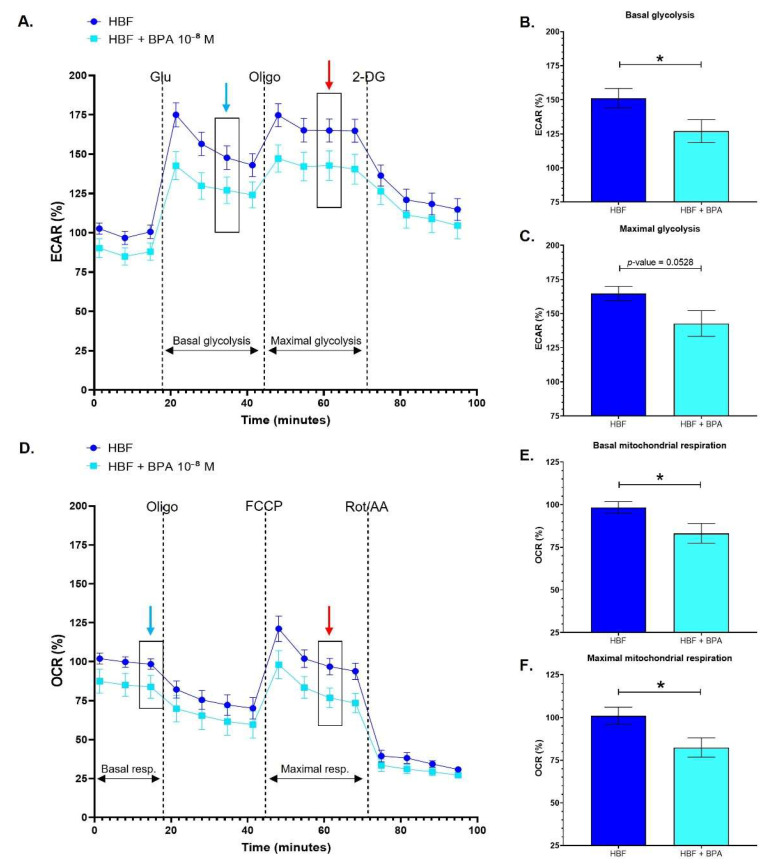
Healthy human bladder fibroblasts exhibit a decreased glycolytic and mitochondrial metabolism following chronic exposure to BPA. (**A**–**C**) ECAR and (**D**–**F**) OCR were determined using the XFe96 Extracellular Flux Analyzer in healthy human bladder fibroblasts exposed (HBF + BPA) or not (HBF) to 10^−8^ M BPA for 8 days to establish (**B**) basal glycolysis, (**C**) maximal glycolytic capacity, (**E**) basal mitochondrial respiration and (**F**) maximal mitochondrial capacity. (**A**) The glycolytic metabolism was established by sequential injections of glucose (Glu), oligomycin (Oligo) and 2-deoxy-glucose (2-DG). Analyses in (**B**,**C**) were performed using measure #6 (blue arrow) for basal glycolysis and measure #10 (red arrow) for maximal glycolytic capacity. (**D**) The mitochondrial respiration was established by sequential injections of oligomycin (Oligo), FCCP and the combination of rotenone (Rot) and antimycin A (AA). Analyses in (**E**,**F**) were performed using measure #3 (blue arrow) for basal mitochondrial respiration and measure #10 (red arrow) for maximal mitochondrial respiration. Data are presented as the mean ± SEM and displayed as percentages of controls (i.e., untreated condition) (*n* = 8; *N* = 3). * *p* < 0.05 by Student’s *t*-test.

**Figure 3 cancers-13-05461-f003:**
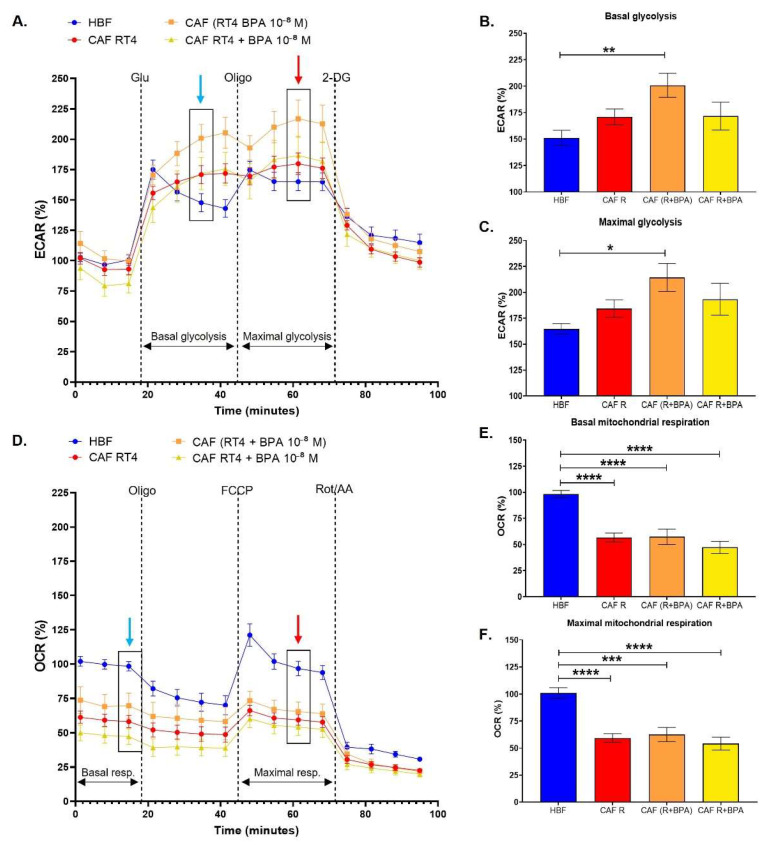
Cancer-associated fibroblasts conditioned by non-invasive bladder cancer cells exhibit a metabolic switch characterized by a decreased mitochondrial metabolism and increased glycolysis accentuated by BPA. (**A**–**C**) ECAR and (**D**–**F**) OCR were determined using the XFe96 Extracellular Flux Analyzer in cancer-associated fibroblasts (CAFs). Healthy human bladder fibroblasts (HBFs) were induced into CAFs over 8 days with conditioned media from non-invasive cancer cells (RT4) with exposure (direct or indirect) or not to 10^−8^ M BPA to establish (**B**) basal glycolysis, (**C**) maximal glycolytic capacity, (**E**) basal mitochondrial respiration and (**F**) maximal mitochondrial capacity. (**A**) The glycolytic metabolism and (**D**) mitochondrial respiration were established using sequential injections, as in Figure 1. Analyses in (**B**,**C**) were performed using measure #6 (blue arrow) for basal glycolysis and measure #10 (red arrow) for maximal glycolytic capacity. Analyses in (**E**,**F**) were performed using measure #3 (blue arrow) for basal mitochondrial respiration and measure #10 (red arrow) for maximal mitochondrial respiration. Data are presented as the mean ± SEM and displayed as percentages of controls (*n* = 8; *N* = 3). * *p* < 0.05, ** *p* < 0.01, *** *p* < 0.001 and **** *p* < 0.0001 by one-way ANOVA.

**Figure 4 cancers-13-05461-f004:**
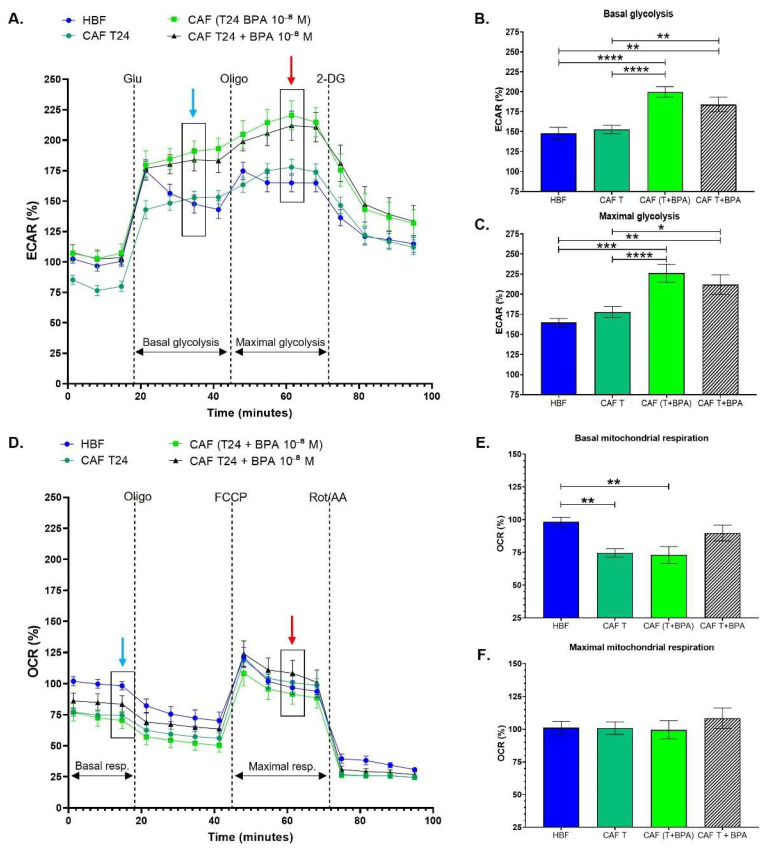
Cancer-associated fibroblasts conditioned with invasive bladder cancer cells in the presence of BPA exhibit an increased glycolytic metabolism. (**A**–**C**) ECAR and (**D**–**F**) OCR were determined using the XFe96 Extracellular Flux Analyzer in cancer-associated fibroblasts (CAFs). Healthy human bladder fibroblasts (HBFs) were induced into CAFs over 8 days with conditioned media from invasive cancer cells (T24) with exposure (direct or indirect) or not to 10^−8^ M BPA to establish (B) basal glycolysis, (**C**) maximal glycolytic capacity, (**E**) basal mitochondrial respiration and (**F**) maximal mitochondrial capacity. (**A**) The glycolytic metabolism and (**D**) mitochondrial respiration were established using sequential injections, as in Figure 1. Analyses in (**B**,**C**) were performed using measure #6 (blue arrow) for basal glycolysis and measure #10 (red arrow) for maximal glycolytic capacity. Analyses in (**E**,**F**) were performed using measure #3 (blue arrow) for basal mitochondrial respiration and measure #10 (red arrow) for maximal mitochondrial respiration. Data are presented as the mean ± SEM and displayed as percentages of controls (*n* = 8; *N* = 3). * *p* < 0.05, ** *p* < 0.01, *** *p* < 0.001 and **** *p* < 0.0001 by one-way ANOVA.

## Data Availability

Data contained within the article.

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
