# Peer review of "Bisphenol A Alters the Energy Metabolism of Stromal Cells and Could Promote Bladder Cancer Progression"

_cancers, 2021, doi:10.3390/cancers13215461_

Round 1
Reviewer 1 Report
This is an interesting research. The aims of the study is important because BPA is , indeed, widely used reagent and bladder as a potential target is definitely might be considered. Interesting the authors opinion about possible endocrine dependency of bladder cancer. however, in this paper this assumption based only on the action of BPA. Would be helpful to look for some particular pathways of BPA action in BC in future. Bladder fibroblasts are, indeed, good model system. However, their transformation to CBF efficiency is not proved and discussed substantially. RT4 and T24 is right selection of non-invasive and invasive BC cell lines. However, nonefficient T24 switch after BPA treatment need further investigation.
In the Materials and methods section culture media described as the same for cell lines and primary cells. I believe that these media are different. Generally this is good well-prepared manuscript with adequately chosen cellular model, and well established research design.
Author Response
Reviewer 1:
- This is an interesting research. The aims of the study is important because BPA is, indeed, widely used reagent and bladder as a potential target is definitely might be considered. Interesting the authors opinion about possible endocrine dependency of bladder cancer. however, in this paper this assumption based only on the action of BPA. Would be helpful to look for some particular pathways of BPA action in BC in future. Bladder fibroblasts are, indeed, good model system. However, their transformation to CBF efficiency is not proved and discussed substantially. RT4 and T24 is right selection of non-invasive and invasive BC cell lines. However, nonefficient T24 switch after BPA treatment need further investigation.
Thank you for your comments. CAF induction was confirmed by FACS analysis during these experiments. We added a sentence in section 2.2 CAF induction to clarify this point: Line 119: “CAF induction was confirmed by α-Smooth muscle actin (α-SMA) expression through FACS analysis, as previously demonstrated [16].”
Ref. 16: Ringuette Goulet, C., et al., Exosomes Induce Fibroblast Differentiation into Cancer-Associated Fibroblasts through TGFβ Signaling. Mol Cancer Res, 2018. 16(7): p. 1196-1204.
Further investigations would indeed be interesting to better understand the mechanisms of BPA effects on bladder cancer. In the present study, our objectives were first to establish the potential links between BPA and bladder cancer, focusing on energy metabolism. We are currently studying the possible pathways of BPA action on bladder cancer.
- In the Materials and methods section culture media described as the same for cell lines and primary cells. I believe that these media are different. Generally this is good well-prepared manuscript with adequately chosen cellular model, and well established research design.
The medium used for cell lines and primary cells is the same, as described in the Materials and Methods, section 2.1 Cell lines. Throughout the study, we used the same medium to make sure the effects observed were due to BPA exposure and not due to media changes.
Reviewer 2 Report
The authors have conducted an important study of bladder cancer stromal cell metabolic changes in response to the endocrine disrupting molecule bisphenol A (BPA). Their studies lay an important framework for future studies to better understand the mechanisms behind and consequences of these alterations in cellular metabolism in response to BPA. The manuscript is very well written, and the review of the literature and discussion present a clear context for the significance of the work. Below, please find a few major points to address to strengthen the scientific interpretation as well as minor points to consider editing for clarity.
Major points:
1) Line 260-265: In the discussion about why HBFs exposed to BPA may demonstrate both decreased glycolytic capacity and mitochondrial respiration, I think the cell detoxification/ABC transporter idea is very interesting. Have ABC transporters been implicated in BPA transport? Perhaps see if this paper or a related paper addresses that point:
https://pubmed.ncbi.nlm.nih.gov/34073890/
“Bisphenol A inhibits the transporter function of the blood-brain barrier by directly interacting with the ABC transporter breast cancer resistance protein (BCRP)” Engdahl et al, 2021
2) In the discussion of fibroblasts needing energy to make and maintain ECM (lines 271-280):
What about growth factor production by these fibroblasts? What is the effect of BPA on the proliferation of HBFs and CAFs in these studies? This data was collected in the CyQuant cell proliferation assay for the normalization of the Seahorse data. Is there a way to present the cell number data, or at least to mention if/how BPA affected rates of cell growth in these models?
3) The differences between the metabolic phenotypes reported from non-invasive cancer cell conditioned medium in Fig. 3 and invasive bladder cancer cells in Fig. 4 are very interesting. Further discussion of what might account for these differences might be helpful to include. TGF beta was cited a few times throughout the manuscript as playing an important role in promoting CAF induction and cancer cell proliferation and migration. What, if anything, is known about TGF beta levels from the invasive vs non-invasive bladder cancer cells, and how might that contribute to the observed metabolic phenotypes? Are there other molecules implicated in these differences? IL-6 from CAFs can act on IL-6R on bladder cancer cells by a paracrine signal—is there a role for CAF autocrine signaling via IL-6 or another signal that could alter cellular metabolism?
4) For the ECAR experiments, why was oligomycin used for injection instead of rotenone/antimycin A? Oligomycin failed to induce a strong increase in ECAR in all cells tested—are these cells already very glycolytic? I was surprised that 2-DG did not completely reduce ECAR back to or below baseline. Could there be, in the absence of rotenone/antimycin A, residual mitochondrial acidification that may account for this observation?
5) Line 163: Could the authors please specify what the mean +/- SEM indicates for each figure? This can be addressed in the figure legends or in Materials and Methods. How many wells or biological replicates? This will help others design similar experiments with HBFs. (For example, in Figure 2, does N=3 mean 3 rounds of HBF isolation from 3 different patients? (Line 194))
Minor points:
Line 60-62: Consider revising the sentence citing reference 12 to include the idea that aerobic glycolysis provides less overall ATP generation (“synthesize ATP faster[, albeit with less overall production,] since…”)
Line 62-63: Consider revising the sentence citing reference 13 by removing “it has been hypothesized that” and please clarify “excess carbons”: perhaps “excess biosynthetic intermediates”?
Line 113: Maybe “exposure” instead of “exposition”? And in Figure 1 legend, line 121 and 125, and in line 210, 224, 226, 247, 277, 307, 319, 352, 353, 370? Apologies if I am missing the purpose of using the word “exposition.”
Line 142: Would it be possible to provide a product number for the oligomycin complex from Cayman? Does the % oligomycin A vary from lot to lot?
Lines 141-150: For the concentrations listed for the injected inhibitors, could the authors please specify if these are the final concentrations in the well?
Line 145: Instead of competitive inhibitor of glucose, could you say “competitive inhibitor of an early step in glycolysis” or something similar?
Line 145: Instead of “established” perhaps “measured” or “tested”?
Line 180: Instead of “reduction in their energy metabolism” perhaps be more specific in this sentence: perhaps something like “decreased glycolytic and mitochondrial oxidative metabolism” (as in Fig. 2 title)?
Line 269-270: Consider specifying a reduced oxidative metabolism in the first sentence (line 269) and decreased energy metabolism when exposed to BPA (line 270)?
Line 310: Consider specifying “By enhancing the glycolytic metabolism of CAFs…”
Line 369-370: Perhaps specify that the energy metabolism of HBFs is negatively affected by BPA?
Line 376: prognosis instead of prognostic?
Line 422: For reference 17, is the bibliography listing of the first author’s name correct? Looks consistent with PubMed records, for instance, but just wanted to check.
Author Response
Reviewer 2:
The authors have conducted an important study of bladder cancer stromal cell metabolic changes in response to the endocrine disrupting molecule bisphenol A (BPA). Their studies lay an important framework for future studies to better understand the mechanisms behind and consequences of these alterations in cellular metabolism in response to BPA. The manuscript is very well written, and the review of the literature and discussion present a clear context for the significance of the work.
We thank the reviewer for his comments.
Below, please find a few major points to address to strengthen the scientific interpretation as well as minor points to consider editing for clarity.
Major points:
- Line 260-265: In the discussion about why HBFs exposed to BPA may demonstrate both decreased glycolytic capacity and mitochondrial respiration, I think the cell detoxification/ABC transporter idea is very interesting. Have ABC transporters been implicated in BPA transport? Perhaps see if this paper or a related paper addresses that point: https://pubmed.ncbi.nlm.nih.gov/34073890/
“Bisphenol A inhibits the transporter function of the blood-brain barrier by directly interacting with the ABC transporter breast cancer resistance protein (BCRP)” Engdahl et al, 2021
The reviewer brings an important point. We have added the following sentence to address this point: Line 272: “In this sense, bisphenols, including BPA, have been shown to interact with the ABC transporter Breast Cancer Resistance Protein (BCRP; coded by the gene ABCG2) [25].”
Ref. 25: Engdahl, E., et al., Bisphenol A Inhibits the Transporter Function of the Blood-Brain Barrier by Directly Interacting with the ABC Transporter Breast Cancer Resistance Protein (BCRP). Int J Mol Sci, 2021. 22(11).
- In the discussion of fibroblasts needing energy to make and maintain ECM (lines 271-280): What about growth factor production by these fibroblasts? What is the effect of BPA on the proliferation of HBFs and CAFs in these studies? This data was collected in the CyQuant cell proliferation assay for the normalization of the Seahorse data. Is there a way to present the cell number data, or at least to mention if/how BPA affected rates of cell growth in these models?
For metabolic analyses, HBFs and CAFs were seeded at 100% confluency at day 1, which keeps their proliferation rate to the minimum throughout treatment and/or induction. Thus, we did not observe any effect of BPA on cell growth or proliferation in these models. Therefore, we did not measure the production of growth factors as it is beyond the scope of this study.
We added the following precision in our manuscript. Line 109: “thus limiting cell proliferation during treatment.”
- The differences between the metabolic phenotypes reported from non-invasive cancer cell conditioned medium in Fig. 3 and invasive bladder cancer cells in Fig. 4 are very interesting. Further discussion of what might account for these differences might be helpful to include. TGF beta was cited a few times throughout the manuscript as playing an important role in promoting CAF induction and cancer cell proliferation and migration. What, if anything, is known about TGF beta levels from the invasive vs non-invasive bladder cancer cells, and how might that contribute to the observed metabolic phenotypes? Are there other molecules implicated in these differences? IL-6 from CAFs can act on IL-6R on bladder cancer cells by a paracrine signal—is there a role for CAF autocrine signaling via IL-6 or another signal that could alter cellular metabolism?
The reviewer raises an interesting point. We added the following sentences to address this point: Line 300: “Furthermore, studies by Ringuette-Goulet et al. have shown that invasive bladder cancer cells (T24) secrete more TGFβ than non-invasive bladder cancer cells (RT4) [16]. Interestingly, TGFβ has been shown to activate the PI3K-Akt-mTOR pathway and promote the accumulation of ATF4 (activating transcription factor 4) in lung fibroblasts, leading to their metabolic reprogramming [36]. Another molecule that could be implicated is IL-6, which could impact the physiological activity of fibroblasts, as well as potentialize the effects of TGFβ [37]. However, whether the alterations mediated by BPA in the energy metabolism of stromal cells are linked to TGFβ or IL-6 remain to be demonstrated.”
Ref. 16: Ringuette Goulet, C., et al., Exosomes Induce Fibroblast Differentiation into Cancer-Associated Fibroblasts through TGFβ Signaling. Mol Cancer Res, 2018. 16(7): p. 1196-1204.
Ref. 36: O'Leary, E.M., et al., TGF-β Promotes Metabolic Reprogramming in Lung Fibroblasts via mTORC1-dependent ATF4 Activation. Am J Respir Cell Mol Biol, 2020. 63(5): p. 601-612.
Ref. 37: Chabaud, S. and V.J. Moulin, Apoptosis modulation as a promising target for treatment of systemic sclerosis. Int J Rheumatol, 2011. 2011: p. 495792.
- For the ECAR experiments, why was oligomycin used for injection instead of rotenone/antimycin A? Oligomycin failed to induce a strong increase in ECAR in all cells tested—are these cells already very glycolytic? I was surprised that 2-DG did not completely reduce ECAR back to or below baseline. Could there be, in the absence of rotenone/antimycin A, residual mitochondrial acidification that may account for this observation?
The addition of rotenone and antimycin A will completely inhibit the mitochondrial electron transport and disrupt the mitochondrial membrane potential, potentially sending apoptosis signals. This situation is not ideal to evaluate the maximal glycolytic capacity. On the contrary, oligomycin inhibits the ATP-coupled oxygen consumption, reducing, but not inhibiting, the electron transport chain, allowing the cell to maximize its glycolysis, with fewer consequences on cell viability. As observed in our results, the injection of oligomycin did not induce a strong increase in ECAR. This suggests that bladder fibroblasts (HBFs) are already very glycolytic cells. In fact, we routinely observe that HBFs acidify the culture media very rapidly, which also suggests high levels of glycolysis. This could explain why oligomycin failed to induce a strong increase in ECAR. As for 2-DG, indeed, it did not completely reduce ECAR back or below baseline, but a significant decrease was still observed, nonetheless. This could suggest non-glycolytic acidification in our model. However, the possibility of this acidification being linked to the mitochondria would need to be demonstrated and is currently beyond the scope of the study.
- Line 163: Could the authors please specify what the mean +/- SEM indicates for each figure? This can be addressed in the figure legends or in Materials and Methods. How many wells or biological replicates? This will help others design similar experiments with HBFs. (For example, in Figure 2, does N=3 mean 3 rounds of HBF isolation from 3 different patients? (Line 194))
The mean +/- SEM (standard error of the mean) represents the variation for the Y-axis variable for each figure. For example, in figure 2B, the mean +/- SEM represents the variation of the basal ECAR.
Each experiment had eight replicates (n = 8), and the experiment was repeated three independent times (N = 3). The following changes were added.
Line 94: “HBFs used were all from the same patient (not transformed primary cell line).”
Line 164: “Each experiment included eight replicates (n = 8), and each experiment was repeated three times (N = 3).”
Minor points:
- Line 60-62: Consider revising the sentence citing reference 12 to include the idea that aerobic glycolysis provides less overall ATP generation (“synthesize ATP faster[, albeit with less overall production,] since…”)
We agree with the reviewer’s comment. We did the following changes: “Aerobic glycolysis allows cancer cells to synthesize ATP faster, although providing less overall ATP generation, since the production of lactate from glucose is much faster than the complete oxidation of glucose through the mitochondria [12].”
- Line 62-63: Consider revising the sentence citing reference 13 by removing “it has been hypothesized that” and please clarify “excess carbons”: perhaps “excess biosynthetic intermediates”?
We agree with the reviewer’s comment. We did the following changes: “Furthermore, the Warburg effect leads to the production of excess biosynthetic intermediates in the cell, allowing it to generate proteins, lipids and nucleotides, therefore supporting the increased proliferation of cancer cells [13].”
- Line 113: Maybe “exposure” instead of “exposition”? And in Figure 1 legend, line 121 and 125, and in line 210, 224, 226, 247, 277, 307, 319, 352, 353, 370? Apologies if I am missing the purpose of using the word “exposition.”
Corrections have been made to replace “exposition” with “exposure”.
- Line 142: Would it be possible to provide a product number for the oligomycin complex from Cayman? Does the % oligomycin A vary from lot to lot?
The product number for the oligomycin complex from Cayman: #11341.
According to the company, the percentage of oligomycin A does not change from lot to lot. We have therefore decided to remove this information from the manuscript.
- Lines 141-150: For the concentrations listed for the injected inhibitors, could the authors please specify if these are the final concentrations in the well?
Yes, the concentrations indicated represent the final concentrations in the wells. The following specification has been added to the text: “The concentrations indicated for each injection represent the final concentrations in the wells.” (Line 151-152)
- Line 145: Instead of competitive inhibitor of glucose, could you say “competitive inhibitor of an early step in glycolysis” or something similar?
We agree with the reviewer’s comment. We did the following changes: “(…) a competitive inhibitor of the first step of glycolysis.”
- Line 145: Instead of “established” perhaps “measured” or “tested”?
We agree with the reviewer’s comment. We did the following changes: “The mitochondrial respiration was measured by the sequential injection of the ATP synthase inhibitor (…)”
- Line 180: Instead of “reduction in their energy metabolism” perhaps be more specific in this sentence: perhaps something like “decreased glycolytic and mitochondrial oxidative metabolism” (as in Fig. 2 title)?
We agree with the reviewer’s comment. We did the following changes: “Therefore, HBFs chronically exposed to physiological concentrations of BPA are characterized by a reduced glycolytic and mitochondrial oxidative metabolism.”
- Line 269-270: Consider specifying a reduced oxidative metabolism in the first sentence (line 269) and decreased energymetabolism when exposed to BPA (line 270)?
Multiple metabolic pathways are affected by BPA, therefore a decreased availability of substrates does not only affect the oxidative metabolism. The following change has been made: “This inhibition could lead to a decreased availability of substrates for metabolic pathways, resulting in a reduced energy metabolism.”
“Our results have shown that HBFs have a decreased energy metabolism when exposed to BPA (…)”
- Line 310: Consider specifying “By enhancingthe glycolytic metabolism of CAFs…”
We agree with the reviewer’s comment. We did the following changes: “Therefore, by enhancing the glycolytic metabolism of CAFs, BPA could accentuate the consequences of lactate in the ECM (…)”
- Line 369-370: Perhaps specify that the energy metabolism of HBFs is negatively affected by BPA?
We agree with the reviewer’s comment. We did the following changes: “Using bladder fibroblasts and CAFs, we demonstrated that the energy metabolism of HBFs is negatively affected by BPA (…)”
- Line 376: prognosis instead of prognostic?
We agree with the reviewer’s comment. We did the following changes: “Therefore, the ubiquitous and continuous exposure to endocrine disruptors, such as BPA, could have an impact on the clinical level and affect patient’s prognosis.”
- Line 422: For reference 17, is the bibliography listing of the first author’s name correct? Looks consistent with PubMed records, for instance, but just wanted to check.
Yes, the first author’s name of reference 17 is correct.
